# Cold Sintering of Li_6.4_La_3_Zr_1.4_Ta_0.6_O_12_/PEO Composite Solid Electrolytes

**DOI:** 10.3390/molecules27196756

**Published:** 2022-10-10

**Authors:** Binlang He, Shenglin Kang, Xuetong Zhao, Jiexin Zhang, Xilin Wang, Yang Yang, Lijun Yang, Ruijin Liao

**Affiliations:** 1State Key Laboratory of Power Transmission Equipment & System Security and New Technology, Chongqing University, Chongqing 400044, China; 2Tsinghua Shenzhen International Graduate School, Tsinghua University, Shenzhen 518055, China

**Keywords:** cold sintering process, composite solid electrolytes, Li_6.4_La_3_Zr_1.4_Ta_0.6_O_12_, polyethylene-oxide, conductivity

## Abstract

Ceramic/polymer composite solid electrolytes integrate the high ionic conductivity of in ceramics and the flexibility of organic polymers. In practice, ceramic/polymer composite solid electrolytes are generally made into thin films rather than sintered into bulk due to processing temperature limitations. In this work, Li_6.4_La_3_Zr_1.4_Ta_0.6_O_12_ (LLZTO)/polyethylene-oxide (PEO) electrolyte containing bis(trifluoromethanesulfonyl)imide (LiTFSI) as the lithium salt was successfully fabricated into bulk pellets via the cold sintering process (CSP). Using CSP, above 80% dense composite electrolyte pellets were obtained, and a high Li-ion conductivity of 2.4 × 10^−4^ S cm^–1^ was achieved at room temperature. This work focuses on the conductivity contributions and microstructural development within the CSP process of composite solid electrolytes. Cold sintering provides an approach for bridging the gap in processing temperatures of ceramics and polymers, thereby enabling high-performance composites for electrochemical systems.

## 1. Introduction

Lithium-ion batteries have been successfully applied for a wide range of implementations, from grid-level energy storage to electric vehicles and personal electronic devices, due to their long cycle life and high charge/discharge rate [1,2]. However, Li-ion batteries that use combinations of organic liquids and lithium salts as electrolytes are flammable and easy to decompose at high temperatures or high voltage [3,4,5]. All-solid-state batteries potentially promise higher energy densities, eliminate safety concerns and provide a broader operating voltage and temperature range compared to conventional liquid Li-ion batteries [6,7,8].

A large number of solid-state electrolytes (SSEs) with high Li-ion conductivity have been investigated in the past decade. SSEs can usually be divided into two major categories, inorganic solid electrolytes (ISEs) and solid polymer electrolytes (SPEs). ISEs mainly include sulfide-based glass/ceramic (Li_10_GeP_2_S_12_ (LGPS) [6]), garnet-type (Li_7_La_3_Zr_2_O_12_ (LLZO) [9,10,11]), NASICON-type (Li_1.3_Al_0.3_Ti_1.7_(PO_4_)_3_ (LATP) [12,13]), perovskite-type (Li_0.34_La_0.56_TiO_3_ (LLTO) [14,15]) and LiPON-type (Li_3_PO_4_) [16], which usually present a high ionic conductivity and an excellent chemical and electrochemical stability [17,18]. However, the high interfacial resistance caused by the loose interfacial contact between these inorganic solid electrolytes and the electrodes is still a big challenge [19]. Recently, a large amount of research has focused on solid polymer electrolytes such as poly(ethylene oxide) (PEO) [20], poly(vinylidene fluoride-hexafluoropropylene) (PVDF-HFP) [21], poly(acrylonitrile) (PAN) [22], poly(methyl methacrylate) (PMMA) [23] and various lithium salts [24,25] (bis(trifluoromethanesulfonyl) imide (LiTFSI) or LiClO_4_, etc.). SPEs generally exhibit good flexibility and high electrode-electrolyte interfacial compatibility; however, they also have serious shortcomings, such as low ionic conductivity (~10^−5^ S cm^−1^ at room temperature) and poor long-term stability, which severely limits their practical applications. The limitations of these two electrolytes are expected to be resolved by combining inorganic electrolytes with flexible polymers to fabricate composite solid electrolytes [26,27].

Garnet-type Li_6.4_La_3_Zr_1.4_Ta_0.6_O_12_ (LLZTO) materials are receiving increasing attention due to their high ionic conductivity (10^−3^ to 10^−4^ S cm^−1^), chemical compatibility with lithium metal, and good mechanical, thermal, and electrochemical stability [28,29]. Poly(ethylene oxide) (PEO) polymer electrolytes were mixed with a lithium salt, such as LiC_2_F_6_NO_4_S_2_ (LiTFSI), to create polymer lithium ion transport channels [30,31]. Furthermore, PEO and LiTFSI can combine with LLZTO to fabricate a ceramic/polymer composite solid electrolyte. However, limited by processing temperature, ceramic/polymer composite solid electrolyte was generally made into thin films rather than sintered into bulk pellets. The traditional sintering temperature of the bulk pellets (~1000 °C) is much higher than that the polymer electrolyte can withstand (<300 °C) [32]. Moreover, the traditional high-temperature sintering process shows obvious deficiencies, including Li loss, impurity phase formation, incompatibility with polymers, and high processing cost [33,34,35]. Hence, advanced sintering technologies are highly desired to prepare low-cost ceramic/polymer composite solid electrolytes without compromising electrochemical/chemical performances.

The cold sintering process (CSP) offers a route to densify ceramics below 300 °C by incorporating a transient solvent phase and uniaxial pressure into the sintering process [36,37,38,39], which allows the co-sintering of ceramics with polymers for applications in SSEs [34]. CSP refers to a multistage non-equilibrium thermodynamic chemo-mechanic process; particle dissolution, mass transport, evaporation of the transient solvent, and precipitation. It provides an approach for bridging the gap between ceramics and polymers, enabling the discovery, design, and fabrication of new ceramic/polymer composite solid electrolytes.

Here, we propose a new route to synthesize ceramic/polymer composite solid electrolytes using the CSP at 120–270 °C. LLZTO-PEOx-LiTFSI composites fabricated at 150 °C demonstrate high relative densities above 80% and conductivities around 10^−4^ S cm^−1^ at room temperature, which is comparable to the LLZTO bulks that were sintered above 1000 °C [40].

## 2. Results and Discussion

In the fabrication of the composite solid electrolytes, the deionized water acts as a liquid phase moistening the interfaces of LLZTO particles, which may induce the dissolution–precipitation process during the heating process of CSP. In addition, PEO-LiTFSI may lubricate the interfaces of LLZTO particles to promote their rearrangement under applied uniaxial pressure, thus creating more connections for mass transport in CSP. The densification patterns of the cold-sintered LLZTO-PEO_x_-LiTFSI composite solid electrolytes are shown in Figure 1a, and the density decreases from 86.9% to 74.1% with increasing PEO content of PEO-LiTFSI. Figure 1b shows the relative density of LLZTO-PEO_2_-LiTFSI cold-sintered at different temperatures. It is noted that the relative densities of the composite electrolytes reach more than 80% from 120 °C to 270 °C. The maximum relative densities (~89%) can be obtained as the cold sintering temperature increases to 270 °C.

The Nyquist plots of LLZTO-PEO_x_-LiTFSI composite electrolytes for evaluating the ionic conductivity are shown in Figure 2a,b. (*R*_grain_*CPE*_grain_)(*R*_gb_*CPE*_gb_)*R*_el_ was fitted as the equivalent circuit for all the Nyquist plots, where *R* and *CPE* are resistance and the constant phase element, and *R*_grai_, *R*_gb_, and *R*_el_ refer to the LLZTO grains, grain boundary, and the electrode, respectively. In Figure 2b, two separate semicircles can be observed in impedance spectra for the LLZTO-PEO_2_-LiTFSI composite electrolytes. The semicircle in the high frequency could correspond to the ion motion in the ceramic grains (*R*_grain_*CPE*_grain_). The semicircle in the low frequency could be attributed to the polymeric grain boundary between the LLZTO ceramic grains (*R*_gb_*CPE*_gb_). The capacitors in the Huggins model were replaced with the constant phase element (*CPE*) to account for any dispersion in the time constants [41,42]. The complex impedance response of a single *CPE* can be given by:(1)$$Z(\omega )=\frac{1}{Q{\left(j\omega \right)}^{n}}$$
where *Z*(*ω*) is the frequency-dependent impedance, *ω* is the frequency, *j* is the imaginary operator, *Q* is a numerical value related to the capacitance, and *n* is the ideality coefficient between 0 and 1. The capacitance values described with the *CPE* can be obtained from Equation (2) [43,44]:(2)$$C={\left(Q\times R^{1-n}\right)}^{1/n}$$
where *C* is the capacitance, and *R* is the resistance. From the individual capacitances of the respective semicircles, the overall capacitance *C*_Total_ can be calculated further using Equation (3) [43]:(3)$$\frac{1}{C_{\mathrm{Total}}}=\frac{1}{C_{\mathrm{grain}}}+\frac{1}{C_{\mathrm{gb}}}$$
where *C*_grain_ and *C*_gb_ represent the capacitances of grains and grain boundary, and the values of capacitance for the cold-sintered LLZTO composite electrolytes are shown in Figure 2c, which fits the reported values for the grain (10^−11^ F) and the grain boundaries (10^−7^ F) quite well [45].

The ionic conductivity (*σ*) is calculated via Equation (4) [46]:(4)σ=L/(R·S)
where *S* and *L* are the area and thickness of the electrolyte, respectively. *R* is obtained by EIS measurement with symmetric cells of the electrolyte sandwiched by two copper electrodes. In Figure 2d, the results show that the composite electrolyte with an EO/Li ratio of 2:1 exhibits an optimal ionic conductivity (2.4 × 10^−4^ S cm^–1^). However, the ionic conductivity of electrolytes decreased as more PEO was added to PEO-LiTFSI, and a similar phenomenon was also presented in previous reports [47,48]. In Figure 2e, the lithium-ion conductivity activation energy was calculated from the slope of the Arrhenius plot using Equation (5):(5)$$\;\sigma =A\exp\left(-E_{\alpha }/kT\right)$$
where *σ* is the conductivity, *A* is the pre-exponential parameter, *E**_α_* is the activation energy, *T* is the absolute temperature, and *k* is the Boltzmann constant. It is found that activation energies of the LLZTO composite electrolytes with varied PEO-LiTFSI are between 0.29 eV and 0.39 eV, which is similar to values of pure or doped LLZTO [49,50,51]. Clearly, among all the samples, LLZTO-PEO_2_-LiTFSI shows the highest room temperature conductivity of 2.4 × 10^−4^ S cm^−1^ with the lowest activation energy of *E**_α_* = 0.29 eV, which indicated that the incorporation of an appropriate PEO_2_-LiTFSI could improve the Li-ion transport in cubic phase garnet LLZTO electrolyte.

Of special interest is that, although the electrolyte density increases with sintering temperature, there is no significant improvement in the ionic conductivity, as shown in Figure 2f. The fabrication of LLZTO-PEO_2_-LiTFSI in CSP, especially at a temperature above 150 °C, may lead to chemical and physical changes in the polymer chains. Linus Froboese [52] proposed that the polymer will be melted and degraded due to the high temperature and excessive mechanical shearing. Additionally, recrystallization of the polymer may occur during CSP and result in a reduction in the amorphous polymer regions, which suppresses the lithium-ion conductivity.

LLZTO-PEO_x_-LiTFSI samples cold-sintered at 150 °C, which present an optimal ionic conductivity, were chosen for microstructure characterization. Figure 3 shows the X-ray diffraction (XRD) patterns of the cold-sintered LLZTO-PEO_x_-LiTFSI (x = 1, 1.5, 2, 5, 8) composite electrolytes. By comparison with the standard card LLZTO (PDF 45-0109), the composite electrolytes show the standard cubic phase with garnet-type structure and good crystallinity, which indicates that the addition of PEO and LiTFSI via CSP cannot change the crystal structure of the LLZTO ceramic. The lattice parameters are extracted from the Rietveld refinement results of XRD patterns for LLZTO-PEOx-LiTFSI, as shown in Appendix A. The lattice constant is about 12.96 Å, similar to the reported values [53,54], and it seems that the lattice constants are not affected by the doped PEOx-LiTFSI during the CSP.

Two reflection peaks of PEO within 18–25° are observable in PEO_x_-LiTFSI, as shown in Figure 3, but the two peaks cannot be observed in ceramic-rich LLZTO-PEO_x_-LiTFSI composites. In addition to these peaks, a small amount of Li_2_CO_3_ was detected in the composite electrolyte because the LLZTO may react with CO_2_ in the air during the electrode’s preparation and the EIS test. Appendix A presents the Raman spectra of the air-exposed LLZTO-PEO_2_-LiTFSI, and characteristic peaks associated with the vibration of C-O-C, O-C-C, and C-C of PEO and the vibration of TFSI^−^ in LiTFSI can be checked at 278.1 and 740.7 cm^−1^, respectively. Li_2_CO_3_ was also found in the Raman spectra; however, after the surface (0.02 mm in thickness) of the sample was polished, only a very small peak related to Li_2_CO_3_ can be observed, indicating that Li_2_CO_3_ on the surface of the sample can be easily removed by polishing.

It is of great importance to optimize the homogeneity of polymers and Li salt in the ceramic/polymer composite solid electrolytes, which are primordial to accurately elaborate the microstructure/properties relationship. Backscattered electron (BSE) images of fracture cross sections of LLZTO-PEO_x_-LiTFSI composite electrolytes with different EO to Li molar ratios are shown in Figure 4a–e. The LLZTO-PEO_1_-LiTFSI sample is porous with worm-like grains and rough grain surfaces in Figure 4a. With a slight increase in molar ratios of EO to Li, the polymer content between the crystals increases, showing a flake-like connection in LLZTO-PEO_1.5_-LiTFSI (Figure 4b). In Figure 4c, it can be found that the polymers are uniformly distributed between grains, forming a conductive bridge that builds a continuous Li^+^ transport pathway through the LLZTO grain and ensures sufficient void space to load the PEO/LiTFSI polymer electrolyte. Seo, J. H. et al. [55] reported that the polymer–salt bridge acting as an ionic transport could be formed at the interfaces between the grains in the highly densified electrode and the solid electrolyte. As the molar ratio of EO to Li continues to increase, excessive PEO content will lead to the segregation of the polymer (Figure 4d,e), which is unfavorable to the transmission of Li^+^ between bulk LLZTO particles. Corresponding to the results of densities, excessive polymer leads to a reduction in the density of the composite electrolyte, which results in larger pores between the particles. Furthermore, Figure 4f shows that there is a slight grain growth with small molar ratios of EO to Li, which can be ascribed to the possible mechanism that Li was preferentially dissolved in the aqueous solution in an incongruent dissolution process and promoted grain growth, similar to the case of LLZO [56]. However, when the value of EO:Li further increases, the high polymer (PEO) content may be aggregated between LLZTO grains, which will block the mass transport and suppress the grain growth [55]. Therefore, a suitable grain size is usually required to ensure stronger grain–grain contacts for improved electrical conductivity.

Figure 5 shows EDS mapping of cold-sintered LLZTO-PEO_2_-LiTFSI composite electrolytes cold-sintered at 150 °C, reflecting the distribution of the elements. The F element from the LiTFSI salt and the C element from the PEO are evenly distributed, as shown in Figure 5b,c, suggesting that the LiTFSI salt is dissolved well in the PEO matrix. It can likewise be seen that the La, Zr, and Ta elements in the LLZTO ceramic are uniformly embedded in the PEO network, and this structure is favorable to aiding lithium-ion transport.

A possible ionic transport mechanism in the cold-sintered LLZTO composite electrolytes is proposed and illustrated in Figure 6. Two possible lithium-ion transport pathways may be available in the composite electrolyte as follows: (1) the lithium-ion transport channel through the inorganic LLZTO electrolyte and (2) the ion conduction along the amorphous polymer grain boundaries between LLZTO grains. The LLZTO bulk can transfer Li ions, while the dissolution of LiTFSI in the PEO aids the lithium ions’ transport through the amorphous area between LLZTO grains. Some other studies pointed out that the amorphous area between ceramic grains plays a major role in improving ion conductivity [57,58]. Lewis proposed that the acid–base interaction between ceramic and anion promotes the dissociation of lithium salt, thus increasing the concentration of mobile lithium ions in the amorphous-rich area to enable fast Li^+^ transport. However, some experimental evidence proves that Li^+^ prefers to move through the garnet phase in PEO-based composite electrolytes [59,60]. Although it is still controversial which pathway is the dominant one for Li^+^ transport, the enhancement of ionic conductivity in LLZTO composite electrolyte can be attributed to the synergy of the two pathways.

## 3. Materials and Methods

### 3.1. Materials Preparation

PEO (*M*_w_ = 5,000,000) and Acetonitrile were purchased from Sigma-Aldrich (Shanghai) Trading Co., Ltd. (Shanghai, China) and dried under a vacuum oven at 50 °C for 24 h. LiTFSI was purchased from Sigma-Aldrich, dried at 100 °C for 12 h, and kept in a glove box filled with Ar atmosphere. The garnet LLZTO was purchased from Hefei Kejing Material Technology Co., Ltd. (Hefei, China). PEO and LiTFSI were dissolved in Acetonitrile at different molar ratios of EO:Li, and the Acetonitrile was stirred using a magnetic stir bar at 75 °C, as shown in Figure 7. The stirring rate was set to 400 r·min^−1^, and the stirring time was 2 h. Afterward, the LLZTO powder was added to the uniformly mixed PEO-LiTFSI solution. The EO to Li molar ratios were set around 1, 1.5, 2, 5, and 8, and the content of LLZTO powders was 90 wt%. Then, the mixture was stirred at 75 °C for 8 h with a rate of 300 r·min^−1^. Finally, the mixture was dried at 100 °C for 12 h to obtain the precursor powder.

### 3.2. Cold Sintering Process

The precursor powder was wetted and ground with deionized water using a mortar and pestle for ten minutes and then transferred into a stainless-steel die (inner diameter of 12.7 mm) with polypropylene (PP) separators between the powder and the punch. The die was then heated using a loop heater, stuck together with a thermocouple, and loaded into a hydraulic press. The mixture powder in the die was cold-sintered at 120–270 °C with a heating rate of 10 °C per minute for 1 h under an assisted uniaxial pressure of 300 MPa via CSP.

### 3.3. Sample Characterization

The density of the sintered pellet was measured by weight and volume, and the Archimedes method in ethanol and the relative density can be calculated with the following equation:*d*_r_ = *ρ*_m_/*ρ*_t_(6)
where *d*_r_ is the relative density of LLZTO-PEO_x_-LiTFSI composite solid electrolytes, and *ρ*_m_ and *ρ*_t_ are the measured density and the theoretical density of the electrolytes, respectively. Structures of the samples were characterized by X-ray diffraction with Cu Kα radiation (Rigaku Ultima IV). The scanning of 2θ angles ranged from 7° to 70° with a step of 0.026°. Morphology and elementary mapping were obtained using a ZEISS Sigma 300 scanning electron microscope (SEM) equipped with an energy dispersive spectrometer (EDS) and a backscattered electron (BSE) detector.

The samples were polished with sandpaper, and Au was sputtered on both sides of the sample as electrodes for electrical measurements. The electrochemical impedance spectroscopy (EIS, Concept 80 Novocontrol) from 25 °C to 300 °C was measured in the frequency range of 0.1 Hz–10 MHz. EIS fitting was conducted using ZView software (Scribner Associates, Southern Pines, NC, USA).

## 4. Conclusions

In summary, LLZTO-PEO_x_-LiTFSI (x = 1, 1.5, 2, 5, and 8) solid electrolytes with different EO to Li molar ratio doping were prepared by CSP at 120–270 °C using deionized water as the auxiliary liquid phase. The effects of different EO to Li molar ratios on the physical phase, microstructure, and electrochemical properties of LLZTO solid electrolytes were investigated. The experimental results show that the relative density of cold-sintered LLZTO-PEO_x_-LiTFSI composite electrolytes can reach about 90%, and the doped PEO and LiTFSI exist in LLZTO as a network structure. The highest ionic electrolyte conductivity above 10^−4^ S cm^−1^ was achieved for LLZTO-PEO_2_-LiTFSI cold-sintered at 150 °C. The incorporated PEO-LiTFSI increases the ionic conductivity and enhances the electrochemical properties of LLZTO. This facile and low-cost CSP method provides a novel route for the fabrication of ceramic/polymer composite electrolytes.

## Figures and Tables

**Figure 1 molecules-27-06756-f001:**
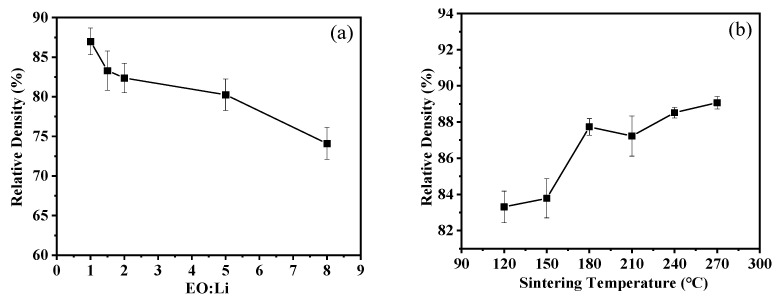
Relative density of (**a**) cold-sintered LLZTO composite electrolytes with different EO to Li molar ratios and (**b**) cold-sintered LLZTO-PEO_2_-LiTFSI composite electrolytes at different sintering temperatures.

**Figure 2 molecules-27-06756-f002:**
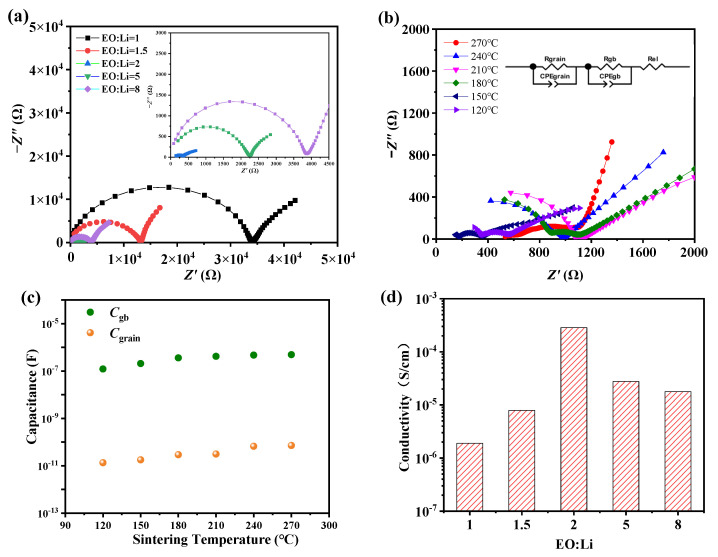
The Nyquist plots of (**a**) cold-sintered LLZTO-PEO_x_-LiTFSI composite electrolytes and (**b**) cold-sintered LLZTO-PEO_2_-LiTFSI composite electrolyte with different sintering temperatures, and the inset shows the equivalent circuit. (**c**) Capacitance versus cold sintering temperatures for the individual *R-CPE* elements of LLZTO-PEO_2_-LiTFSI. (**d**) The ionic conductivity of cold-sintered LLZTO-PEO_x_-LiTFSI composite electrolytes. (**e**) Arrhenius curves of cold-sintered LLZTO composites with various EO:Li. (**f**) Cold-sintered LLZTO-PEO_2_-LiTFSI composite electrolyte with different sintering temperatures.

**Figure 3 molecules-27-06756-f003:**
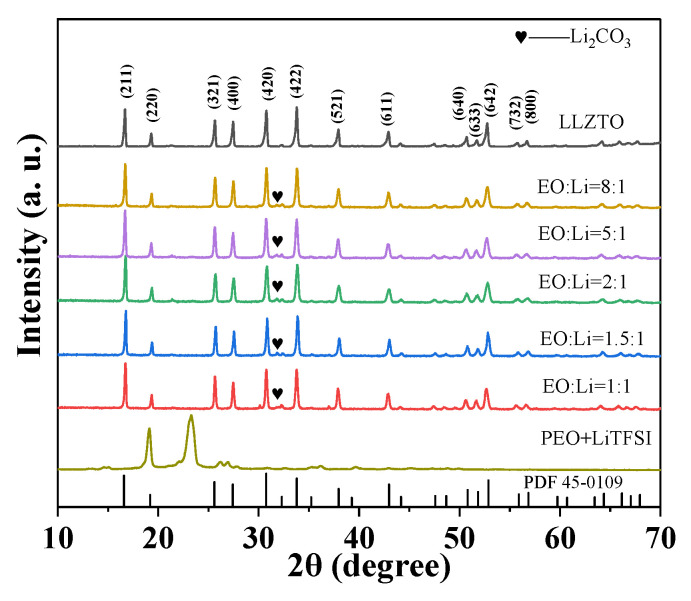
XRD patterns of different EO to Li molar ratios for cold-sintered LLZTO composite electrolytes.

**Figure 4 molecules-27-06756-f004:**
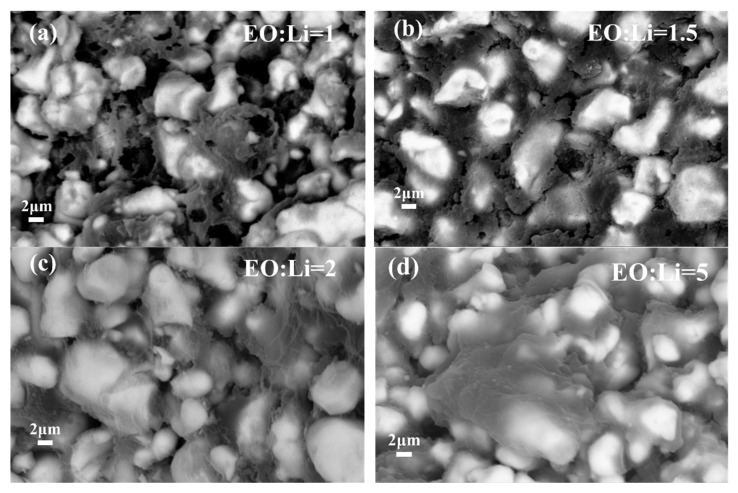
(**a**–**e**) BSE images and (**f**) grain size of cold-sintered LLZTO-PEO_x_-LiTFSI composite electrolytes with different EO to Li molar ratios.

**Figure 5 molecules-27-06756-f005:**
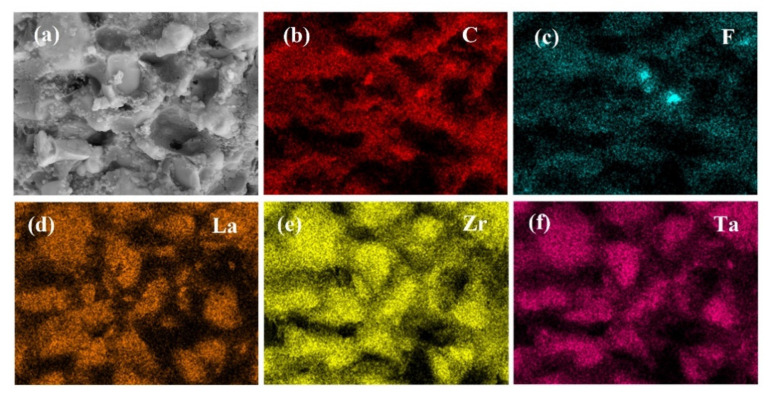
(**a**) SEM images and (**b**–**f**) EDS mapping results of the best performance cold-sintered LLZTO-PEO_2_-LiTFSI composite electrolyte.

**Figure 6 molecules-27-06756-f006:**
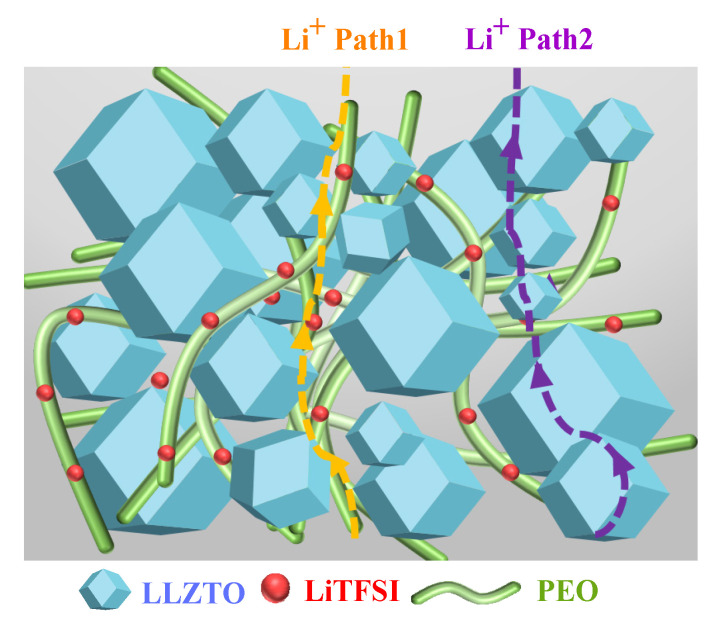
Schematic illustration of Li^+^ transport mechanism in the cold-sintered LLZTO-PEO_x_-LiTFSI composite electrolytes.

**Figure 7 molecules-27-06756-f007:**
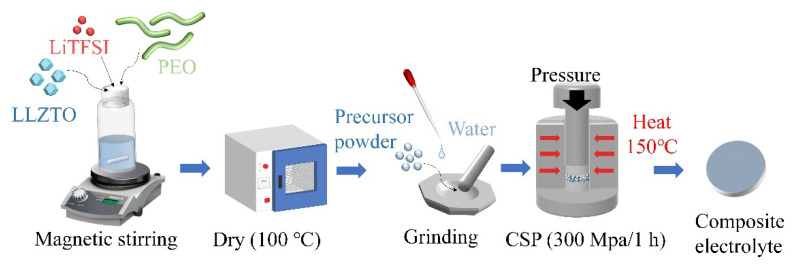
Schematic illustration of cold sintering to produce composite electrolytes comprised of ceramics and polymers.

## Data Availability

Data are available upon reasonable request.

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
