# Peer review of "Cold Sintering of Li6.4La3Zr1.4Ta0.6O12/PEO Composite Solid Electrolytes"

_molecules, 2022, doi:10.3390/molecules27196756_

Round 1
Reviewer 1 Report
The paper deals with the preparation and characterization of composite electrolyte (CSP) made of Li6.4La3Zr1.4Ta0.6O12 (LLZTO) particles dispersed in polyethylene oxide (PEO). The topic is highly relevant in the field of batteries as composite electrolytes made of Li-ion conducting ceramics mixed with polymer electrolyte are regarded as the potential next electrolyte for high energy densities batteries.
The CSPs addressed her is ceramic rich with 90 wt.% and heat-pressed as a pellet for characterization. Typical and appropriate characterization techniques, such as XRD, SEM-BSE and EIS, are used throughout the study.
However, the approach though is not necessarily new. Numerous papers already deal with ceramic particles dispersed in a polymer matrix. In addition, many approximations are made in the document and in the experimental procedures. No strong analysis of the experimental findings is neither provided in the paper. On these bases, the paper as such cannot be accepted in Molecules Journal.
Some of the reasons for such decision are further detailed in the following:
1 – Figure 1 describing the experimental procedure used for preparing the pellets does not match the cold sintering process explained in the text. Water is included during a grinding steps after drying and before cold sintering according to Figure 1 while it is not mentioned in the text. Water (and air exposition) is highly detrimental in contact with garnet ceramics as the particles surface react. LITFSI salt used in PEO matrix is also highly hygroscopic and will influence the resulting conductivities. No further attention is brought on the residual water in the materials or no explanation on the removal of water is explained.
2 – The pellet is expected to be ceramic-rich in order to reach cold sintering. 90 wt.% of ceramic is used with the balance being PEO-LITFSI polymer. The wt.% density of LLZO is within 5 g/cm3 when it is around 1.3 for Polymer. This means that 90 wt.% of ceramic represents lower vol.%. Consequently, the materials are polymer rich and may not be considered as good candidates for cold sintering process.
3 - The densification of the pellets is a key factor in ceramic electrolyte explaining the conductivity of the materials. The calculation for the density of the pellets is not explained. Is it based solely on the theoretical density of LLZTO ceramic or on the Archimedes method is here used in ethanol. In the presence of a polar solvent, LITFSI will dissolved in it. Consequently, the density based on the Archimedes method may be biased. As more salt will be found in polymers with lower EO:Li, perhaps, this is one of the reasons why the density is found to decrease with increasing EO:Li?
In Figure 3b, it is shown that the density of the pellet increases with the sintering temperature. No indication of which pellet composition is provided. Is it a pellet with high EO:Li or low EO:Li?
4 – A sintering temperature in the range of 120 to 300C is used. In this range, PEO will melt at ca. 180C favoring which may explain the density jump at this temperature. At more elevated temperature, PEO may degrade. Not to mention LITFSI salt thermal behavior. Still no explanation is provided regarding these aspects.
5 – As a ceramic, LLZTO required high sintering temperature in order to induce grain growth and brindging of the particles, usually around 1000C. In the present case, with temperatures lower than 300C, grain growth would be highly surprising. Instead of grain growth with varying of EO:Li, as more polymer is found with higher EO:Li ratio and considering the vol.% of polymer in the materials, are the “larger grains” rather ascribed as high polymer containing materials embedding ceramic particles?
6 – The ionic conductivity is obviously temperature dependent. This dependence should be shown to explain the conductivity mechanisms based on the activation energies. A conduction mechanism is proposed based on 2 RC circuits. One could be ascribed to the ceramic part, the other to the polymer one. The authors highlight the presence of an interface and its key character for the ionic conduction. No RC circuit is used to explain this part. Still no attempt to explain the conductivities measured here using 3 RC circuits.
Author Response
Dear Editor and reviewers,
Thank you for taking time to review and consider our paper entitled “Cold sintering of Li6.4La3Zr1.4Ta0.6O12/PEO composite solid electrolyte” for consideration by Molecules. We would like to thank the reviewers for thoroughly reviewing our manuscript and making many thoughtful comments. We were very pleased to see that all reviewers recognized the novelty and potential significance of our work.
We have added significant new data, and addressed the reviewers’ comments in the attached response letter (a pdf file). The changes made in light of reviewers’ comments are marked in red in the revised manuscript for your convenience. We believe our paper is now ready for publication. Please let us know how we can proceed. Thank you for your time and consideration.
We are looking forward to hearing from you.
Yours sincerely,
Dr. Xuetong Zhao
E-mail: zxt201314@cqu.edu.cn
State Key Laboratory of Power Equipment & System Security and New Technology, Chongqing University, Chongqing, 400044, P. R. China.

Reviewer 2 Report
The article titled ”Cold sintering of Li6.4La3Zr1.4Ta0.6O12/PEO composite solid electrolyte” by He et al. focuses on the fabrication method for the synthesis of composite electrolytes via cold sintering process. The manuscript is well written, however I feel the authors should test their electrolyte within the full cell configuration before being considered for publication within this journal. It is quite usual for the research community to just report on conductivities without really testing their electrolytes within the full cells and see the cycling behavior. Further, the composite electrolytes not only should provide good ionic conductivities but also good mechanical properties, which the authors may consider. Other than this I having small points with regards to the manuscript:
1. In the experimental section, the molecular weight of the PEO used should be mentioned.
2. In the line 124 the authors mention “Co-doping of PEO and LiTSFI cannot change the structure of LLZTO”. I think sentence should be rephrased since by definition the use of doping is not justified here.
3. In the line 125 to 127, authors mention formation of Li2CO3 due to the reaction of garnet with air, which is well known. This can also contribute to the interfacial resistances within the composite electrolytes. Is there a reason that handing of garnet powders was not done under inert atmosphere?
4. Authors show the XRD patterns of composite electrolytes but do not show the one from PEO+LiTSFI. This must be shown. PEO usually shows reflections within the 18 degrees to 25 degrees range, which cannot be observed in ceramic-rich composites. However, one can see an increased background or an amorphous bump within this range which is not the case here. Can the authors throw some light on this?
5. From Line 171 through to 173, the authors mention that well developed interfaces can be lead to reduced interfacial reference. However, this is not exactly true (https://doi.org/10.1021/acsami.1c05846) due to the reaction of the lithium salt with the garnet. Authors should clear this within the manuscript.
6. The authors needs to provide more depth to the impedance discussion. The authors use two R-CPE elements but do not mention the capacitances found. The authors also mention lithium loss in line 197 with cold sintering. What is the origin of such a lithium loss, is this from the garnet or the lithium salt?
7. The authors show possible ionic transport mechanism without having any experimental proof of this within their manuscript. From the impedance only two transport processes based on bulk and grain boundary are derived, then how can the authors come to the transport mechanism conclusion?
Author Response
Dear Editor and reviewers,
Thank you for taking time to review and consider our paper entitled “Cold sintering of Li6.4La3Zr1.4Ta0.6O12/PEO composite solid electrolyte” for consideration by Molecules. We would like to thank the reviewers for thoroughly reviewing our manuscript and making many thoughtful comments. We were very pleased to see that all reviewers recognized the novelty and potential significance of our work.
We have added significant new data, and addressed the reviewers’ comments in the attached file of "response letter". The changes made in light of reviewers’ comments are marked in red in the revised manuscript for your convenience. We believe our paper is now ready for publication. Please let us know how we can proceed. Thank you for your time and consideration.
We are looking forward to hearing from you.
Yours sincerely,
Dr. Xuetong Zhao
E-mail: zxt201314@cqu.edu.cn
State Key Laboratory of Power Equipment & System Security and New Technology, Chongqing University, Chongqing, 400044, P. R. China.

Reviewer 3 Report
I am asking the authors for the following changes:
Fig. 2 - XRD patterns of composites - please show lattice constants and Miller indicies for all materials as well as for pure LLZTO. Which card number is correct PDF 80-0457 or PDF 45-0109 for LLZTO ?
Fig. 4 - How to explain the increase of composites grain size (the composite EO:Li 2 - the largest size) and then their decrease size ?
Fig. 6 - How is the greatest conductivity of the composite (EO:Li = 2) explained ? Is this conductivity related to the grain size of material ?
The visualization of pure LLZTO structure is welcome.
Author Response

(The authors gave the same response as above.)

Round 2
Reviewer 2 Report
I see the manuscript has been improved significantly and the points raised during last review have been addressed.
Author Response
Thanks for the reviewer's accept for the revised manuscript.
Reviewer 3 Report
I accept corrected manuscript for publication in Materials.
Author Response

(The authors gave the same response as above.)
